# Effects of an Intermittent Fasting 5:2 Plus Program on Body Weight in Chinese Adults with Overweight or Obesity: A Pilot Study

**DOI:** 10.3390/nu14224734

**Published:** 2022-11-09

**Authors:** Junren Kang, Xiaodong Shi, Ji Fu, Hailong Li, Enling Ma, Wei Chen

**Affiliations:** Department of Clinical Nutrition, Peking Union Medical College Hospital, Chinese Academy of Medical Sciences and Peking Union Medical College, Shuai Fu Yuan Dongcheng District, Beijing 100730, China

**Keywords:** obesity, weight loss, intermittent fasting, calorie restriction, meal replacement

## Abstract

To retrospectively review the efficacy of intermittent fasting 5:2 plus program (30% of energy requirements on fast days and 70% of energy requirements on nonfasting days) in Chinese patients with overweight or obesity. This retrospective cohort study evaluated weight loss outcomes of patients treated with 12 weeks weight loss program in clinic. Adults with overweight or obesity participated in intermittent fasting 5:2 plus, daily calorie restriction (70% of energy requirements every day) or daily calorie restriction with meal replacement (70% of energy requirements every day, partly provided with high-protein meal replacement) programs for 12 weeks. The primary objective was to compare the weight loss of three groups. The primary outcome measure was the change in the % total weight loss. A total of 131 patients (35.3 ± 10.1 years; 81.7% female) were included, and the mean weight loss was 7.8 ± 4.4% after 12 weeks. The intermittent fasting 5:2 plus group (*n* = 42) achieved 9.0 ± 5.3% weight loss, compared with 5.7 ± 3.7% in the daily calorie restriction group (*n* = 41) (*p* < 0.001) and 8.6 ± 3.5% in the meal replacement group (*n* = 48) (*p* = 0.650). A total of 85.7% of patients in the intermittent fasting 5:2 plus group lost more than 5% body weight, vs. 58.5% in the daily calorie restriction group (*p* = 0.008), and 45.2% lost more than 10% body weight, vs. 14.6% in the daily calorie restriction group (*p* = 0.005). No serious adverse events were reported in the three groups. The intermittent fasting 5:2 plus program more effectively led to weight loss than daily calorie restriction in the short-term in Chinese patients with overweight or obesity. A longer-term prospective randomized controlled trial is needed.

## 1. Introduction

The worldwide obesity prevalence has increased significantly, and China has the largest population of obesity [1,2]. Intensive nutrition and lifestyle modifications are basic, effective and safe interventions [3,4,5]. Intermittent fasting (IF) is a common intensive nutrition and lifestyle intervention strategy for obesity [6,7].

The mechanisms of IF in weight loss and metabolic health are complex and include enhancing defenses against oxidative and metabolic stress, eliminating or remediating damaged molecules and improving insulin resistance [8,9,10]. IF has been widely used in many long-term follow-up and large sample studies as a reliable intervention for obesity [11,12,13]. Common protocols of IF consist of time-restricted feeding [14], alternate-day fasting [15] and intermittent fasting 5:2 [12,13,16] diets.

Currently, in clinical practice IF programs provide 30–40% of energy requirements on fasting days and 100% or ~125% of energy requirements on nonfasting days [12,15]. A previous study found that more than 100% of energy requirements on nonfasting days increased fasting insulin [17], therefore, studies with less than 100% of energy requirements on nonfasting days were designed. In a 16-week RCT, an alternate-day fasting program providing approximately 1200 kcal of energy on nonfasting days led to improvements in fasting triglycerides and insulin and weight loss [18]. Meal replacements were used for energy restriction in this study, and the meal replacement programs for weight loss were simple, novel and effective [19,20,21].

Energy restriction on nonfasting days with alternate-day fasting model was shown to be effective in previous study [18]. However, research regarding energy restriction on nonfasting days with other IF models was limited. It is unclear whether energy restriction without meal replacements on nonfasting days would be more effective for weight loss, especially in the intermittent fasting 5:2 model. We have exploratory applied an intermittent fasting 5:2 plus program in which approximately 30% of energy requirements were provided on fasting days (two nonconsecutive days a week) and 70% of total energy requirements were provided on nonfasting days. To validate the efficacy of the intermittent fasting 5:2 plus program for weight loss, a retrospective cohort study was conducted to compare the effects of the intermittent fasting 5:2 plus program and two continuous energy restriction approaches for weight loss in Chinese patients with overweight or obesity.

## 2. Materials and Methods

### 2.1. Patient Characteristics

This was a retrospective analysis of 12-week weight loss data. Data from electronic database of patients with overweight or obesity receiving 12 weeks of weight loss intervention at the Peking Union Medical College Hospital (PUMCH) from 1 July 2018 to 1 July 2020 were reviewed. The patients with overweight or obesity receiving different weight loss interventions were included. (1) Intermittent fasting 5:2 plus group: the patients were provided approximately 30% of their energy requirements on fasting days (two non-consecutive days a week) and 70% of their total energy requirements on nonfasting days. (2) Daily calorie restriction (CR) group: the patients were provided approximately 70% of their total energy requirements per day. The targeted percentages of energy derived from fat, protein, and carbohydrates were 20%, 20%, and 60%, respectively. The average eating frequency was defined as 5 times a day, including breakfast, lunch, dinner and two extra meals with fruit or yogurt at 10 AM and 4 PM. (3) Daily calorie restriction with high protein meal replacement (HP) group: the patients were provided approximately 70% of their total energy requirements per day. The targeted percentages of energy derived from fat, protein, and carbohydrates were 30%, 30%, and 40%, respectively. Approximately 50% of protein was provided by whey protein isolate powder as a meal replacement for breakfast and a snack in 4 PM. One scoop (23.2 g) of whey protein a meal replacement contained of 88.8 kcal, 20.0 g protein, 0.5 g fat, 0.6 g carbohydrate and 0.084 g sodium (manufactured by Milk Specialties Global, Ltd., Wautoma, WI, USA). These patients were asked to complete 150–300 min of moderate-intensity aerobic or resistance exercise per week. And several behavioral strategies were also advised and consisted of reducing high-fat food consumption, having regular eating times, having an average eating frequency of 5 times a day, eating vegetables and protein first and carbohydrates later, increasing the number of chewing cycles, self-weighing every day, avoiding prolonged periods of sitting, exercising every day and going to sleep early. Data on weight and body compositions were collected when the patients were reviewed in clinic every 4 weeks. Dietary education was provided by trained registered dietitians or physicians in clinic.

The inclusion criteria were as follows: (1) 18–70 years old; (2) BMI > 25 kg/m^2^, and (3) a 12-week weight loss intervention. Patients were excluded if they could not complete the 12-week weight loss intervention. The patients with type 1 or 2 diabetes, cardiovascular events, or cancer were excluded. The study was approved by the Ethics Committee of PUMCH (approval No. S-K 780).

### 2.2. Outcomes

The primary objective was to compare the weight loss effects of three groups for 12 weeks of weight loss. The primary outcome measure was the change in the % total weight loss (TWL). The secondary outcome measures were the fat mass (FM): total mass (TM) ratio and the fat free mass (FFM): total mass (TM) ratio. The body weights, body mass index (BMI) and body compositions of patients were measured after fasting for 8 h at 0 and 12 weeks with multi-frequency segmental bioelectrical impedance analysis (BIA, H-Key 350, SEEHIGHER, CHINA). Weight loss was defined as the initial weight minus the final weight. The % TWL was defined as the difference between the initial weight and final weight divided by the initial weight.

### 2.3. Data Collection

Clinical data such as age, sex, weight and body compositions at 0 and 12 weeks were collected from an independent electronic database. Data on adverse events were collected from the patients’ self-reported complications. The serious adverse events were define as death, pancreatitis, cholecystitis, serious abdominal pain of unknown aetiology. Data were abstracted and inputted independently by 2 trained investigators to ensure consistency and integrity. Disagreements on information were resolved by consensus or by retrieving further information from the electronic database.

### 2.4. Statistical Analysis

For the sample size calculation, IF was shown to reduce body weight by 6% in previous papers [15]. We calculated that approximately 29 patients would provide 95% power to detect a significant difference of 6% between baseline and week 12 (two-sided a = 0.05, β = 0.1). Thus, we aimed to recruit 39 patients per group with an expected dropout rate of 25%. The differences among three groups were compared by one-way ANOVA or Mann-Whitney U test and the chi-square test or Fisher’s exact test for dichotomous or categorical variables. The differences within groups between baseline and 12 weeks later were analyzed using paired *t* test for normally distributed variables or Wilcoxon test for nonnormally distributed continuous variables. A two-tailed *p* value of less than 0.05 was considered statistically significant. Statistical analyses were performed with SPSS software (Version 19, SPSS Inc., IBM, Armonk, NY, USA).

## 3. Results

### 3.1. Patient Characteristics

From 1 July 2018 to 1 July 2020, 531 patients with overweight or obesity received weight loss treatment in the same nutritional clinic. A total of 10 patients were excluded for out of age, 294 patients receiving less than 12 weeks treatment were excluded, 96 patients with type 1 or 2 diabetes, cardiovascular events, or cancer were excluded. A total of 131 patients with overweight or obesity who met the inclusion criteria were included. A flowchart of the recruitment was given in Figure 1. The mean age of the patients was 35.3 ± 10.1 (range, 19 to 67 years), and 81.7% (*n* = 107) were female. The mean body weight was 84.8 ± 16.4 kg, with a mean BMI of 30.7 ± 4.7 kg/m^2^. The baseline characteristics of patients were comparable among three groups (Table 1).

### 3.2. The Differences among Groups

At week 12, the weight change from baseline was 7.9 ± 5.0 kg in the IF group, 7.5 ± 3.6 kg in the HP group and 4.7 ± 3.4 kg in the CR group (Variance analysis, *p* = 0.001) (Table 2). The % TWL from baseline was 9.0 ± 5.3% in the IF group, 8.6 ± 3.5% in the HP group and 5.7 ± 3.7% in the CR group, and as shown in Figure 2, the % TWL among three groups had significant differences (ANOVA, *p* < 0.001).

There were also significant differences in percentages of more than 5% weight loss and more than 10% weight loss among groups (Fisher’s exact test, *p* = 0.008 and 0.005, respectively) (Table 2). Compared to the CR group, proportion of subjects with more than 5% weight loss and more than 10% weight loss were higher in the IF and HP groups. Approximately 85.7% of patients in the IF group lost more than 5% body weight and 81.3% of patients in the HP group, while 45.2% of patients in the IF group and 41.7% in the HP group lost more than 10% body weight. There was no significant differences in the % TWL (Fisher’s exact test, *p* = 0.650), proportion of subjects with more than 5% weight loss (Fisher’s exact test, *p* = 0.57) and more than 10% weight loss (Fisher’s exact test, *p* = 0.730) between the IF group and HP group.

### 3.3. The Differences within Groups

After 12 weeks of weight loss intervention, 131 patients lost an average of 6.8 ± 4.3 kg and had a 7.8 ± 4.4% reduction in body weight. The changes in weight, BMI, FM/TM and FFM/TM were shown in Table 3. Forty-two patients were in the intermittent fasting 5:2 plus group, the weight change at week 12 from baseline was 7.9 ± 5.0 kg (Paired *t* tests for differences within groups, *t* = 10.203, *p* < 0.001). 41 were in the daily calorie restriction group, the weight change from baseline was 4.7 ± 3.4 kg (Paired *t* tests, *t* = 8.890, *p* < 0.001). 48 were in the daily calorie restriction with high protein meal replacement group, the weight change from baseline was 7.5 ± 3.6 kg (Paired *t* tests, *t* = 14.390, *p* < 0.001). As shown in Figure 3, the BMI and FM/TM of the three groups were all significantly decreased at week 12 compared to baseline, whereas FFM/TM was increased.

### 3.4. Adverse Events

During follow-up, all patients were interviewed face-to-face every 4 weeks. No serious adverse events were reported in the three groups. The common self-reported complications were constipation (6.8%), fatigue (5.3%) and hair loss (4.6%). There were no significant differences in the rates of constipation, fatigue and hair loss among the three groups (Fisher’s exact test, *p* = 0.257, 0.597 and 0.985, respectively).

## 4. Discussion

In the current study, we demonstrated that the intermittent fasting 5:2 plus program produced superior weight loss compared with the daily calorie restriction for 12 weeks in Chinese patients with overweight or obesity. There were no serious adverse events during the intermittent fasting 5:2 plus program.

The worldwide obesity prevalence has increased significantly. People with obesity have a higher risk of diabetes, hyperlipidemia, hyperuricemia, nonalcoholic fatty liver, polycystic ovary syndrome, kidney disease, and cancer, consuming many medical insurance costs and national public health investments [1,2]. As a common intensive nutrition and lifestyle intervention strategy for obesity, IF has been shown to achieve weight reduction and improve insulin resistance and metabolic health [6,22,23]. Intermittent fasting has been widely used in individuals with obesity, cardiovascular disease [24] and diabetes mellitus [25,26]. The reliability of IF has been verified in long-term studies [11,12,13].

Several types of IF have been reported [27]. Time-restricted feeding [28,29], alternate-day fasting [30,31] and intermittent fasting 5:2 [12,13,20] are common types of IF. Alternate-day fasting and intermittent fasting 5:2 programs provide 30–40% of energy on fasting days and 100% or more of energy needs on nonfasting days [12,15].

The provision of less than 100% of energy requirements on nonfasting days has been explored in RCTs. An alternate-day fasting study provided approximately 70% of energy on nonfasting days led to improvements in fasting triglycerides and insulin and weight loss [18]. In this intermittent fasting 5:2 study, 30% of energy requirements on fasting days and 70% on nonfasting days were provided. The % TWL was higher in the intermittent fasting 5:2 group than in the daily calorie restriction group, which was in line with the results of a previous study [18]. Calorie restriction has been shown to improve insulin resistance, lower blood lipids, fasting insulin and reduce inflammation [5,32,33,34].

Both the CR and HP groups had equal percentages of caloric restriction (−30% of total energy requirements), and the % TWL was higher in the HP group than in the CR group. This finding is consistent with the outcomes of previous studies. Meal replacement programs for weight loss were associated with eliminating choices, controlling portions and enhancing adherence [19,20,21]. On the other hand, meal replacements may require a higher overall cost and are associated with adverse effects in 76.8% of individuals [21], and individuals only partially replacing meals with meal replacements might quickly regain weight [35]. In this study, the meal replacement group had less weight loss at 12 weeks than the IF group, although the difference was not significant.

Therefore, in addition to reduced calorie intake, other mechanisms may underlie the weight loss observed with IF. Several mechanisms of IF have been reported, such as metabolic switching, metabolic adaptations, enhanced mitochondrial health and blunted inflammation [8,10,36]. IF may also influence the intestinal microbiota and human circadian rhythms, resulting in impacts on obesity [37].

In previous studies, the common adverse effects of IF included dizziness, headache, nausea and temporary sleep disturbances [12]. Participants in IF programs sometimes felt hungry and irritable on fasting days, which usually improved within a month [8]. In this study, a daily multivitamin supplement and 2000–2500 mL water were recommended to prevent the risk of complications, and regular face-to-face interviews were recommended to monitor complications. Behavioral interventions, such as going to sleep early [38,39], eating an average frequency of 5 times a day [40], avoiding prolonged periods of sitting [41] and eating vegetables and protein first [42], were also advised to develop healthy habits. There were no serious adverse events, which was in consistent with a previous literature [16]. In a secondary analysis of dietary data of two studies of intermittent fasting 5:2 in New Zealand, Scholtens EL et al. found that although fibre intake of IF had lower than recommended and IF was safe and acceptable for weight-loss [16].

There are several limitations in our study. First, this was a retrospective cohort study, and the study group assignment was not randomized, although the baseline characteristics were comparable among groups. A prospective randomized controlled trial is needed in the future. Second, this is a short-term study and the duration of 12 weeks was relatively short. The effects of weight loss were superior in 8-week or 12-week time-restricted feeding interventions compared with CR interventions [14,43], but were without obvious advantages in the 12-month weight loss program [15,44]. Studies of the long-term follow-up should be conducted in the future [45]. Third, the patients with BMI > 25 kg/m^2^ were enrolled in this study, and we cannot generalize the benefits of the 5:2 plus program in other populations. For example, in lean, healthy adults, intermittent fasting was shown to less effectively improve postprandial indices of cardiometabolic health and reduce body fat mass than CR in a randomized controlled trial [46].

Nonetheless, to our knowledge, this is the first exploratory study to examine the weight loss effect of 12-week intermittent fasting 5:2 plus intervention in Chinese patients with overweight or obesity. The findings of this study could help adults with overweight or obesity select an appropriate intervention for short-term weight loss.

## 5. Conclusions

The intermittent fasting 5:2 plus program led to effective weight loss in the short-term in Chinese patients with overweight or obesity. A longer-term prospective randomized controlled trial is needed.

## Figures and Tables

**Figure 1 nutrients-14-04734-f001:**
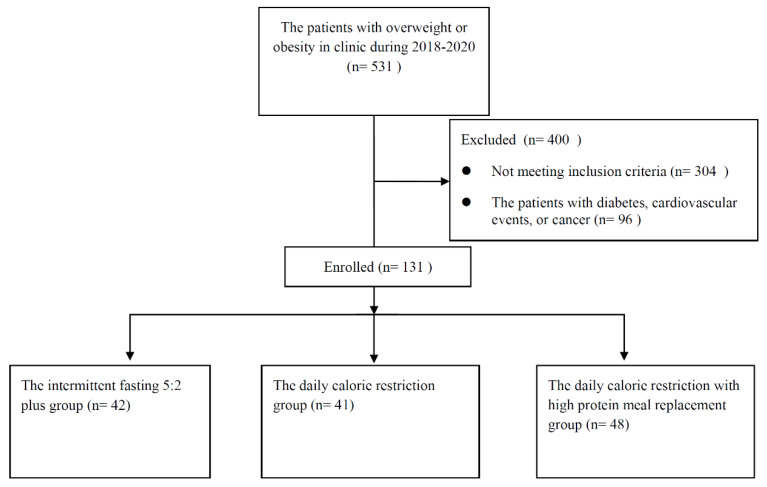
Study Flow Diagram.

**Figure 2 nutrients-14-04734-f002:**
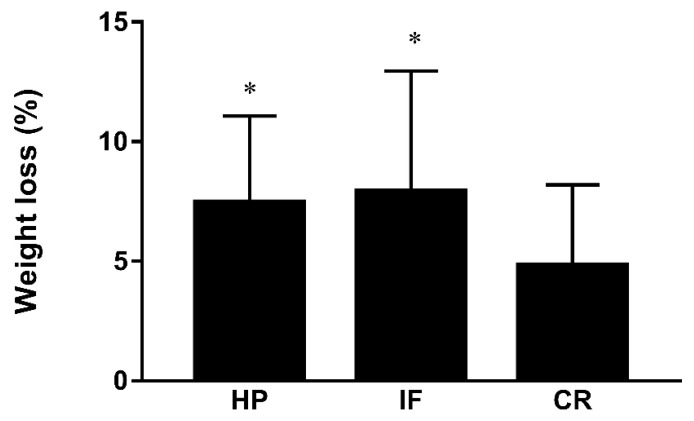
The differences in % total weight loss between groups. IF, Intermittent fasting 5:2 plus; CR, daily calorie restriction; HP, high protein with portion meal replacements. * *p* < 0.001 for difference among the three groups.

**Figure 3 nutrients-14-04734-f003:**
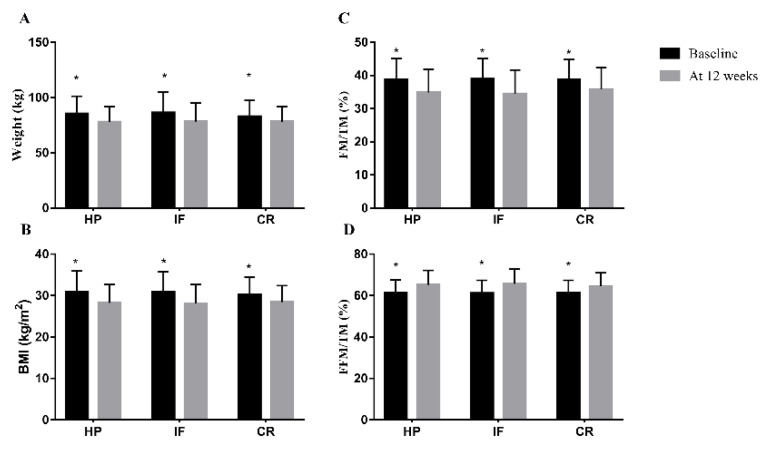
The differences within groups. (**A**) Weight; (**B**) BMI; (**C**) FM/TM; (**D**) FFM/TM. BMI, Body Mass Index; IF, Intermittent fasting 5:2 plus; CR, daily calorie restriction; HP, high protein with portion meal replacements; FM, fat mass; TM, total mass; FFM, fat free mass. * *p* < 0.001 for difference from baseline in groups.

**Table 1 nutrients-14-04734-t001:** Baseline participant characteristics.

Characteristics	IF (*n* = 42)	CR (*n* = 41)	HP (*n* = 48)	*p* for Difference among Groups
Age, y, mean ± SD	34.7 ± 9.8	37.5 ± 11.7	34.0 ± 8.9	0.25
Range	22–63	22–67	19–58	
Sex (%)				0.472
Male	9 (21.4)	5 (12.2)	10 (20.8)	
Female	33 (78.6)	36 (87.8)	38 (79.2)	
Height, cm	166.1 ± 9.3	165.4 ± 7.4	166.1 ± 7.4	0.896
Weight, kg	86.2 ± 18.6	82.8 ± 14.6	85.3 ± 16.0	0.624
BMI, kg/m^2^	30.9 ± 4.9	30.2 ± 4.2	30.8 ± 5.1	0.772
Basic energy expenditure, kcal	1529.3 ± 298.0	1463.3 ± 156.7	1492.7 ± 196.6	0.473

SD, Standard deviation; BMI, Body Mass Index; IF, Intermittent fasting 5:2 plus; CR, daily calorie restriction; HP, high protein with portion meal replacements.

**Table 2 nutrients-14-04734-t002:** Participant characteristics after 12 weeks of weight loss intervention.

Characteristics	IF (*n* = 42)	CR (*n* = 41)	HP (*n* = 48)	*p* for Difference among Groups
Weight loss, kg, mean ± SD	7.9 ± 5.0	4.7 ± 3.4	7.5 ± 3.6	0.001
Percentage of weight loss, %	9.0 ± 5.3	5.7 ± 3.7	8.6 ± 3.5	<0.001
5% weight loss (%)	36 (85.7)	24 (58.5)	39 (81.3)	0.008
10% weight loss (%)	19 (45.2)	6 (14.6)	20 (41.7)	0.005

SD, Standard deviation; BMI, Body Mass Index; IF, Intermittent fasting 5:2 plus; CR, daily calorie restriction; HP, high protein with portion meal replacements.

**Table 3 nutrients-14-04734-t003:** Change in study outcomes at baseline and 12 weeks.

Characteristics	IF (*n* = 42)	CR (*n* = 41)	HP (*n* = 48)	*p* for Difference among Groups
Weight, kg, mean ± SD				
Baseline	86.2 ± 18.6	82.8 ± 14.6	85.3 ± 16.0	0.624
Week 12	78.2 ± 16.9 *	78.1 ± 13.6 *	77.8 ± 14.2 *	0.99
BMI, kg/m^2^				
Baseline	30.9 ± 4.9	30.2 ± 4.2	30.8 ± 5.1	0.772
Week 12	28.1 ± 4.6 *	28.4 ± 4.0 *	28.2 ± 4.5 *	0.934
FM/TM, %				
Baseline	38.9 ± 6.3	38.7 ± 6.1	38.8 ± 6.3	0.99
Week 12	34.5 ± 7.2 *	35.7 ± 6.7 *	34.9 ± 7.0 *	0.706
FFM/TM, %				
Baseline	61.1 ± 6.2	61.3 ± 6.2	61.2 ± 6.4	0.994
Week 12	65.6 ± 7.2 *	64.3 ± 6.6 *	65.2 ± 7.0 *	0.71

SD, Standard deviation; BMI, Body Mass Index; IF, Intermittent fasting 5:2 plus; CR, daily calorie restriction; HP, high protein with portion meal replacements; FM, fat mass; TM, total mass; FFM, fat free mass. * *p* < 0.001 for difference from baseline in groups.

## Data Availability

The data that support the findings of this study are included in this article. Further inquiries can be directed to the corresponding author if needed.

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
