# Peer review of "Effects of an Intermittent Fasting 5:2 Plus Program on Body Weight in Chinese Adults with Overweight or Obesity: A Pilot Study"

_nutrients, 2022, doi:10.3390/nu14224734_

Round 1

Reviewer 1 Report

To review the efficacy of intermittent fasting 5:2 plus program, in this work adults with overweight or obesity participating in intermittent fasting 5:2 plus, daily calorie restriction (70% of energy requirements every day) or daily calorie restriction with meal replacement (70% of energy requirements every day, partly provided with high-protein meal replacement) programs for 12 weeks.

The manuscript presents a sufficiently detailed methodology and an indication of correctly used statistical analyses.

The results are well presented and discussed, indicating merit of the manuscript.

However, the Introduction section does not make clear what this work presents in relation to other studies already published. 

This should be made explicit in the introduction.

Minor revisions.

- The abstract must clearly indicate the objective of the work.

Author Response

Replies to reviewers and editors:

We would like to thank the reviewers for their constructive and positive comments.

Reviewer #1

To review the efficacy of intermittent fasting 5:2 plus program, in this work adults with overweight or obesity participating in intermittent fasting 5:2 plus, daily calorie restriction (70% of energy requirements every day) or daily calorie restriction with meal replacement (70% of energy requirements every day, partly provided with high-protein meal replacement) programs for 12 weeks.

Q1: The manuscript presents a sufficiently detailed methodology and an indication of correctly used statistical analyses. The results are well presented and discussed, indicating merit of the manuscript. However, the Introduction section does not make clear what this work presents in relation to other studies already published. This should be made explicit in the introduction.

Response 1:

Thank you very much for your professional comments.

We are sorry for unclear description.

In previous study, energy restriction on nonfasting days with alternate-day fasting model was shown to be effective [1]. However, research regarding energy restriction on nonfasting days with other IF models was limited. Common protocols of IF consist of alternate-day fasting [1, 2] and intermittent fasting 5:2 [3] models. It is unclear whether energy restriction without meal replacements on nonfasting days would be more effective for weight loss, especially in the intermittent fasting 5:2 model.

The revised section is as followed.

“Energy restriction on nonfasting days with alternate-day fasting model was shown to be effective in previous study. However, research regarding energy restriction on nonfasting days with other IF models was limited. It is unclear whether energy restriction without meal replacements on nonfasting days would be more effective for weight loss, especially in the intermittent fasting 5:2 model.”

REFERENCES

  1. Bowen J, Brindal E, James-Martin G, Noakes M. Randomized Trial of a High Protein, Partial Meal Replacement Program with or without Alternate Day Fasting: Similar Effects on Weight Loss, Retention Status, Nutritional, Metabolic, and Behavioral Outcomes. Nutrients. 2018 Aug 23;10:1145. doi: 10.3390/nu10091145. PMID: 30142886; PMCID: PMC6165084.
  2. Trepanowski JF, Kroeger CM, Barnosky A, Klempel MC, Bhutani S, Hoddy KK, et al. Effect of Alternate-Day Fasting on Weight Loss, Weight Maintenance, and Cardioprotection Among Metabolically Healthy Obese Adults: A Randomized Clinical Trial. JAMA Intern Med. 2017 Jul 1;177:930-938. doi: 10.1001/jamainternmed.2017.0936. PMID: 28459931; PMCID: PMC5680777.
  3. Scholtens EL, Krebs JD, Corley BT, Hall RM. Intermittent fasting 5:2 diet: What is the macronutrient and micronutrient intake and composition? Clin Nutr. 2020 Nov;39:3354-3360. doi: 10.1016/j.clnu.2020.02.022. Epub 2020 Feb 22. PMID: 32199696.

Minor revisions.

Q2: - The abstract must clearly indicate the objective of the work.

Response 2:

Thank you very much for your professional comments.

The objective had been added.

“The primary objective was to compare the weight loss of three groups.”

Reviewer 2 Report

This is an excellent and well-conducted study. Congratulations!

The Main question addressed by the research is How a program with intermittent fasting is changing the body composition and weight of the patients.

The topic is original and is an emerging strategy for changing body composition, and weight, controlling glycemia levels, and lipidic metabolism.   It is a contribution to the current knowledge, exploring 5:2 intermittent fasting Versus high protein diet and calories restriction diet, most pf the studies so far didn’t include those 3 arma in comparison.   The conclusions are consistent with the evidence and arguments presented and they address the main. The Reference are appropriate .

Author Response

Thank you very much for your professional comments.